# Learn from Known Unknowns: A Unified Empirical Bayesian Framework for Improving Group Robustness

## Abstract

The lack of group robustness has emerged as a critical concern in machine learning, as conventional methods like Empirical Risk Minimization (ERM) can achieve high overall accuracy while yielding low worst-group accuracy in minority groups. This issue often stems from spurious correlations—non-essential features that models exploit as shortcuts—which can compromise deep learning models in high-stakes applications. Previous works have found that simply retraining classifiers with reweighted datasets or rebalanced samples could significantly improve robustness. However, existing methods lack a unified framework, as they often exhibit inconsistent performance across datasets, and sometimes rely heavily on hyperparameter tuning, making them impractical for real-world datasets. In this work, we first argue that existing methods can be unified as one Empirical Bayesian framework, where a priori of group information is not specified. We then propose our method *Learn from Known Unknowns* under this framework by quantifying the epistemic uncertainty of biased ERM models and introducing a selective reweighting technique for retraining. Our empirical results demonstrate that this approach improves group robustness across diverse datasets and reduces reliance on hyperparameter tuning, offering a more efficient and scalable solution to spurious correlations.

## 1 Introduction

Machine learning models are notoriously sensitive to spurious correlations—brittle associations between prediction targets and non-essential features of the input, such as background, texture, or secondary objects (Geirhos et al., 2019; Stock & Cisse, 2018; Baker et al., 2018; Beery et al., 2018; Sagawa et al., 2020). These correlations can lead to models achieving high average accuracy but failing on specific groups, especially when those groups differ from the majority in ways that the model has incorrectly learned to associate with the target label. This problem is generally described as group robustness in machine learning. A prominent example is in the healthcare domain, where convolutional neural networks trained for pneumonia detection mistakenly relied on hospital-specific metal tokens in chest X-rays rather than focusing on the pathological features of the disease (Zech et al., 2018). This problem is exacerbated by imbalanced group representation in the training data and the use of standard Empirical Risk Minimization (ERM) (Vapnik, 1999), which can cause models to be overly dependent on these spurious features as predictive shortcuts. As a result, model robustness and interpretability are severely compromised in critical, real-world applications.

To mitigate the effects of spurious correlations, a plethora of approaches have been developed. It has been shown that simply retraining classifiers with a reweighted dataset or learned failures from a biased classifier can significantly improve group robustness (Liu et al., 2021; Nam et al., 2020a; Yao et al., 2022). A notable line of work is *Deep Feature Reweighting* (DFR) (Kirichenko et al., 2023; Izmailov et al., 2022) and its variants like SELF (LaBonte et al., 2024b), which achieve state-of-the-art group robustness by retraining the last layer of a model using a group-balanced held-out dataset. While effective in reducing worst-group errors, these methods still face significant practical challenges. Specifically, obtaining accurate group labels requires expert knowledge and labor-intensive annotations, which is infeasible for large-scale datasets. Therefore, most recent works (Qiu et al.,

2023; Li et al., 2024) have moved the focus to improving group robustness without any group annotations. However, class-balancing techniques during fine-tuning have shown inconsistent performance across datasets (LaBonte et al., 2024a), highlighting the urgent need for a better understanding of this problem and more adaptive methods that learn the latent group information by themselves.

To address these challenges, we propose to unify existing methods under an Empirical Bayes framework, which provides a principled approach to estimate and adjust for unknown group biases (Robbins, 1992; Efron, 2024). In this framework, we estimate its beliefs on group information as latent variables based on observed training data. Methods like JTT (Liu et al., 2021), LfF (Nam et al., 2020a), LISA (Yao et al., 2022), and SELF (LaBonte et al., 2024b) can be interpreted within this framework, where they attempt to mitigate spurious correlations by indirectly inferring group labels and reweighting the data. However, because the prior distributions over group assignments are not specified or heuristically chosen, these methods may not achieve the optimal solution under the framework. They often rely heavily on hyperparameter tuning and exhibit inconsistent performance across different datasets, indicating that they do not fully exploit the posterior distribution over group assignments. This underscores the limitations of current approaches within the Empirical Bayes framework and highlights the need for a more principled method that can effectively utilize posterior information to improve robustness.

Motivated by this, we explore the integration of uncertainty quantification into the proposed Empirical Bayes framework to achieve a near-optimal solution under it. Uncertainty quantification, particularly epistemic uncertainty, provides valuable information about the model's confidence in its predictions due to limited knowledge or data (Kendall & Gal, 2017; Hüllermeier & Waegeman, 2021). By quantifying epistemic uncertainty, we can better identify instances where the model is likely relying on spurious correlations, which can inform a more effective reweighting strategy. Recent advances in evidential deep learning offer scalable methods for estimating epistemic uncertainty without the need for specifying priors (Sensoy et al., 2018; Amini et al., 2020), making them well-suited for our approach within the Empirical Bayes framework.

To this end, we build on these insights by leveraging epistemic uncertainty to address the limitations of current group robustness techniques. Specifically, we propose a novel selective reweighting technique that quantifies the epistemic uncertainty of biased ERM models and uses this information to create a self-adaptive held-out set for retraining. Our approach not only enhances model robustness across various datasets but also reduces reliance on hyperparameter tuning, offering a more scalable and generalizable solution to addressing spurious correlations.

**In summary, we make the following contributions:**

- We unify current popular group robustness methods under the Empirical Bayes framework, where the group information is not given or from a held-out set.

- Based on this framework, we propose the method *Learn from Known Unknowns* that achieves optimal performance by inferring the posterior distribution through epistemic uncertainty quantification on the ERM model.

- We demonstrate through empirical results that our method improves robustness across diverse datasets using the uncertainty-informed group estimation and reduces dependence on hyperparameter tuning.

## 2 RELATED WORKS

**Group Robustness Methods.** Group robustness methods focus on training models that perform consistently across predefined groups within a dataset, especially when some groups are underrepresented or prone to spurious correlations. Traditional methods like Group Distributionally Robust Optimization (Group DRO) (Sagawa et al., 2020) utilize explicit group annotations to minimize worst-case group loss, while Deep Feature Reweighting (DFR) (Kirichenko et al., 2023) retrains the final layer using a group-balanced dataset to reduce reliance on spurious features. In scenarios without group labels, approaches such as Invariant Risk Minimization (IRM) (Arjovsky et al., 2019) and Learning from Failure (LfF) (Nam et al., 2020a) attempt to learn invariant representations or use failure patterns to mitigate spurious correlations. However, these methods often depend on strong assumptions or heuristics that may not generalize well, highlighting the need for adaptive techniques that can construct reliable fine-tuning sets without group annotations (LaBonte et al., 2024a). Our

work addresses this gap by leveraging epistemic uncertainty to identify samples affected by spurious correlations, enabling a self-adaptive retraining process.

**Empirical Bayes Methods.** Empirical Bayes (EB) methods combine frequentist and Bayesian ideas by estimating priors directly from data, allowing for adaptive inference without a fully specified prior (Robbins, 1992). Widely used in areas like shrinkage estimation (James & Stein, 1992) and false discovery rates (Efron, 2024), EB methods excel in settings where the distribution of some random variables are not obtainable and latent. Thus, it must be inferred from observations. One example would be the James-Stein estimator (James & Stein, 1992) which exemplifies the EB principle by shrinking estimates toward a common mean to reduce variance. In group robustness, EB provides a natural framework for learning group-specific parameters without explicit group labels. Methods like JTT (Liu et al., 2021) and LfF (Nam et al., 2020a) implicitly use empirical Bayes-like principles, updating group information based on model failures. Our work formally gives an Empirical Bayes Framework to summarize previous methods and extends this perspective by leveraging epistemic uncertainty to guide group-balanced training set reconstruction, improving robustness to spurious correlations across groups without relying on additional annotations.

**Uncertainty Quantification.** Uncertainty quantification is crucial for assessing the reliability of deep learning models, particularly in safety-critical applications. It is typically divided into aleatoric uncertainty, arising from inherent data noise, and epistemic uncertainty, stemming from limited knowledge about model parameters (Kendall & Gal, 2017). Methods like Bayesian Neural Networks (Blundell et al., 2015), Monte Carlo Dropout (Gal & Ghahramani, 2016), and Deep Ensembles (Lakshminarayanan et al., 2017) estimate uncertainty with multiple model parameters and can be computationally intensive. Evidential deep learning (Sensoy et al., 2018) offers an efficient yet effective alternative by modeling evidence as parameters of a Dirichlet distribution, capturing both uncertainties without the need for sampling or ensembles. While uncertainty quantification has been applied to model calibration, active learning, and out-of-distribution detection, its use in improving group robustness is less explored. Building on these insights, our method employs epistemic uncertainty to selectively reweight samples during retraining, reducing the model's reliance on spurious features without requiring explicit group labels or extensive hyperparameter tuning.

## 3 EMPIRICAL BAYESIAN FRAMEWORK

### 3.1 PRELIMINARIES

We consider classification problems with training data points $(x, y) \in \mathcal{X} \times \mathcal{Y}$, where the data is divided into multiple *groups* (subpopulations) $g \in \mathcal{G}$. These groups $g = (y, a)$ are typically characterized by a tuple of a label $y \in \mathcal{Y}$ and a spurious attribute $a \in \mathcal{A}$. The attribute $a$ represents features that are predictive but non-essential to the label and are spuriously correlated with $y$ in the training set. These correlations, however, may not hold in the test set. Each data point $(x, y)$ belongs to one group $g \in \mathcal{G}$. Given the training dataset $\mathcal{D}_{\text{train}} = \{(x_1, y_1), \ldots, (x_n, y_n)\}$ with $n$ samples. Our goal is to learn a model $f_{\theta^*} : \mathcal{X} \to \mathcal{Y}$, parameterized by $\theta^* \in \Theta$, that achieves good performances across all groups on the test set $\mathcal{D}_{\text{test}}$. The standard approach to training classification models is using Empirical Risk Minimization (ERM): given a loss function $\ell$, find the model $\theta$ that minimizes the average training loss (Sagawa et al., 2020):

$$\hat{\mathcal{L}}_{\text{ERM}}(f_\theta; x, y) = \min_{\theta \in \Theta} \frac{1}{n} \sum_{i=1}^{n} \ell(f_\theta(x_i), y_i). \tag{1}$$

While ERM-trained models can achieve high overall accuracy, they tend to rely heavily on spurious features and underperform on minority groups, evaluated by the worst-group accuracy (WGA). It is defined as the minimum accuracy across groups of test samples, i.e.,

$$\text{WGA}(f_\theta; x, y) := \min_{g \in \mathcal{G}} \mathbb{E}_{(x,y) \sim \mathcal{D}_g}[\ell_{0-1}(x, y; \theta)], \tag{2}$$

where $\mathcal{D}_g$ is the data distribution for group $g$ and $\ell_{0-1}(x, y; \theta) = \mathbf{1}[f_\theta(x) \neq y]$ is the 0-1 loss. We will use it as the main metric to evaluate the performance of group robustness.

## 3.2 An Empirical Bayesian Perspective of Group Robustness

Assume we already have the ERM models with parameter $\theta$ as discussed. It has been shown that despite the reliance on unknown spurious features $a$, ERM models can still learn high-quality representations of the core features (Kirichenko et al., 2023). Thus, improving the biased ERM models with better estimated group knowledge (informative prior) could break the spurious correlations and improve the group robustness of the models. In this section, we formulate this problem from an Empirical Bayesian perspective.

In the group robustness problem, we model the generation of data points $(x, y)$ using latent group variables $g \in \mathcal{G}$, which are imbalanced and unknown. These latent groups reflect the challenges of group robustness, as certain groups might be overrepresented (majority) or underrepresented (minority) in the training data, contributing to the spurious bias in the model. The data generation process is described as follows:

$$g \sim p(g), \quad (x, y) \sim p(x, y|g), \tag{3}$$

where $p(g)$ is the prior distribution over groups, and $p(x, y|g)$ is the likelihood of observing the data point $(x, y)$ given group $g$.

We compute the predictive distribution $p(y|x, \theta)$ through the marginalization of group $g$:

$$p(y|x, \theta) = \int_g p(g|x, \theta)p(y|x, \theta, g)dg \tag{4}$$

Given the probability of trained ERM models $p(\theta)$, the main objective of improving group robustness is to find an updated model $\theta^*$ from the ERM models such that:

$$\theta^* = \underset{\theta}{\operatorname{argmax}}\, p(\theta|y, x) = \underset{\theta}{\operatorname{argmax}} \frac{p(y|x, \theta)p(\theta)}{p(y)} \tag{5}$$

$$= \underset{\theta}{\operatorname{argmax}} \frac{p(\theta) \cdot \int p(g|x, \theta)p(y|x, \theta, g)\, dg}{p(y)} \tag{6}$$

$$\propto \underset{\theta}{\operatorname{argmax}}\, p(\theta) \cdot \int p(g|x, \theta)p(y|x, \theta, g)\, dg \tag{7}$$

However, we have no prior knowledge of the group information $p(g|x, \theta)$ and the distribution on the model weight $p(\theta)$. In practice, $\hat{p}(\theta)$ can be estimated by multiple runs of optimization, while there is no direct way to estimate $\hat{p}(g|x, \theta)$. Thus, the estimation of $\hat{p}(g|x, \theta)$ will decide how well the model can learn robust group representation.

## 3.3 Theoretical Guarantee for Empirical Bayes Estimation

In this section, we show that using the observations (e.g. the prediction results) from the trained model can infer a good estimation of $\hat{p}(g|x, \theta)$ under exponential family, inspired by Tweedie's formula (Efron, 2011) in the context of statistics.

**Theorem 3.1.** *Assume that the data generation process is that each data point $(x, y)$ is associated with a latent group variable $g \in \mathcal{G}$, where $g \sim p(g)$ and $(x, y) \sim p(x, y|g)$ under **exponential family**. Suppose that:*

*1. The marginal likelihood $p(y|x, \theta)$ is differentiable with respect to $y$.*
*2. The conditional distribution $p(y|x, g, \theta)$ belongs to an exponential family.*
*3. The variance $\sigma^2$ of the conditional distribution $p(y|x, g, \theta)$ can be estimated.*

*The posterior mean of the group variable $g$ given $x$, $y$, and $\theta$ can be estimated as:*

$$\mathbb{E}[g|x, y, \theta] \approx \mathbb{E}[g] + \sigma^2 \frac{\partial}{\partial y} \log p(y|x, \theta), \tag{8}$$

*where $\mathbb{E}[g]$ is the prior mean of the group variable. This estimation allows us to infer $\hat{p}(g|x, y, \theta)$ from the observed behavior of the model $f_\theta$. The detailed proof is provided in Appendix D.*

| Method | $\hat{p}(\theta)$ | $\hat{p}(g \mid x, \theta)$ | Remark |
|--------|-------------------|-----------------------------|--------|
| LfF | Minimize weighted loss using another biased model to calculate the weight: $$\hat{p}(\theta) \propto \exp\left( - \sum_i w_i\, \ell(f_\theta(x_i), y_i) \right)$$ The weight $w_i$ is calculated by the relative difficulty score between the biased and debiased models. | Estimate minority probability using the prediction from another biased ERM: $\hat{p}(g \mid x, \theta_{\text{bias}}) \propto \ell_{\text{bias}}(x, y)$ | Samples with high loss under the biased model are likely from the minority group. The debiased model upweights the loss for these samples. |
| JTT | Identify the error set: $\mathcal{M} = \{i \mid f_\theta(x_i) \neq y_i\}$ Retrain on unweighted misclassified samples: $$\hat{p}(\theta) \propto \exp\left( - \sum_{i \in \mathcal{M}} \lambda_{\text{up}} \ell(f_\theta(x_i), y_i) \right)$$ | Indicated by misclassified samples from the ERM model: $\hat{p}(g \mid x, \theta_{\text{ERM}}) = \mathbb{I}[f_{\theta_{\text{ERM}}}(x) \neq y]$ | Misclassified samples are treated as minority groups. Retraining on these samples improves robustness without explicit group labels. |
| CnC | Train the encoder using a supervised contrastive loss: $\hat{p}(\theta) \propto$ $$\exp\left( - \sum_i \ell(f_\theta(x_i), y_i) + \lambda \mathcal{L}_{\text{contrast}}(\theta) \right)$$ | Identify groups by clustering learned representations from contrastive learning: $\hat{p}(g \mid x, \theta) = \text{Cluster}(f_\theta(x))$ | Learn group-invariant representations using contrastive learning. Cluster features from the representation to infer groups. |
| DFR | Retrain the last layer using weights derived from a group-balanced validation set: $\hat{p}(\theta_{\text{last}}) \propto$ $$\exp\left( - \sum_i v_{g_i}\, \ell(f_{\theta_{\text{last}}}(x_i), y_i) \right)$$ | Use ground truth group labels from the validation set: $\hat{p}(g \mid x, \theta) = \delta(g = g_i)$ | Group annotations from the validation set are used to retrain the last layer, correcting for bias by balancing group contributions. |
| SELF | Retrain the last layer using either misclassified samples or those with model disagreement: $$\hat{p}(\theta) \propto \exp\left( - \sum_{i \in D} w_i\, \ell(f_\theta(x_i), y_i) \right)$$ , where $D = \operatorname{argmax}_{S \subseteq X, |S|=n} \sum_{x \in S} c(f(x), g(x))$ | Infer group labels through disagreement: $\hat{p}(g \mid x, \theta) \propto$ Disagreement$(f_\theta(x), g_\theta(x))$ is measured by KL divergence. | Use one-half of the validation data with group label group labels with the early-stop disagreement criterion for selecting retraining samples. Fine-tune the last layer based on disagreement. |

Table 1: A brief discussion on how existing methods can be integrated into the Empirical Bayesian framework in terms of $\hat{p}(\theta)$ and $\hat{p}(g \mid x, \theta)$. Refer to the original papers for detailed implementations.

## 3.4 Previous Methods under the Framework

Previous representative methods (Nam et al., 2020a; Liu et al., 2021; Zhang et al., 2022; Kirichenko et al., 2023; LaBonte et al., 2024b) follow this paradigm where each method tries to use different approaches to estimate $\hat{p}(g|x, \theta)$ and optimize $\hat{p}(\theta)$. In Table 1, we show the implementation of each approach on the estimation, which aligns with the non-parametric Empirical Bayes methods. These methods demonstrate that estimating $\hat{p}(g|x, \theta)$ enables models to adjust training procedures to improve group robustness without explicit group labels.

## 4 Learn from Known Unknowns

Building upon the insights from the Empirical Bayesian framework presented in the previous section, we explore a probabilistic approach to estimate $\hat{p}(g \mid x, \theta)$ instead of relying solely on rule-based methods. To this end, we propose a novel method, *Learn from Known Unknowns*, which leverages model uncertainty to estimate latent group probabilities and addresses selection biases inherent in training data. We then utilize these uncertainty-informed empirical priors to retrain the ERM model under the Bayesian Model Averaging (BMA) through reweighting. By adopting an Empirical Bayesian perspective, the proposed method improves robustness across all groups without requiring explicit group labels.

### 4.1 Evidential Second-order Risk Minimization

First, we employ evidential deep learning (Sensoy et al., 2018; Ulmer et al., 2021) as a proxy to estimate the probability $p(y|x, \theta, g)$. The predictions with high uncertainty indicate that the corresponding sample is more probable to the minority groups. The trained evidential ERM model can capture both the prediction and the epistemic uncertainty, which is crucial for empirically inferring latent group probabilities. For a classification task with $K$ classes, the model outputs non-negative

evidence values $e_k(x) \geq 0$ for each class $k$. The parameters of the Dirichlet distribution are defined as:

$$\alpha_k(x) = e_k(x) + 1, \quad \text{for } k = 1, \ldots, K.$$

The Dirichlet distribution over the class probabilities $\mathbf{p}(x) = [p_1(x), p_2(x), \ldots, p_K(x)]$ is given by:

$$\text{Dir}(\mathbf{p}(x) \mid \boldsymbol{\alpha}(x)) = \frac{1}{B(\boldsymbol{\alpha}(x))} \prod_{k=1}^{K} p_k(x)^{\alpha_k(x)-1},$$

where $B(\boldsymbol{\alpha}(x))$ is the multivariate Beta function. The expected class probabilities are:

$$\mathbb{E}[p_k(x)] = \frac{\alpha_k(x)}{S(x)}, \quad \text{where } S(x) = \sum_{k=1}^{K} \alpha_k(x).$$

We formulate the loss function as:

$$\mathcal{L}(x_i, y_i) = \underbrace{-\log(\mathbb{E}[p_{y_i}(x_i)])}_{\text{Classification Objective}} + \lambda \cdot \underbrace{\text{KL}(\text{Dir}(\boldsymbol{\alpha}(x_i)) \| \text{Dir}(\mathbf{1}))}_{\text{Evidence Regularization}},$$

where $\lambda$ is a regularization coefficient and $\text{Dir}(\mathbf{1})$ represents a uniform Dirichlet prior with all concentration parameters equal to 1. The Evidence Regularization term penalizes overconfidence and promotes uncertainty where appropriate. We define it using the Kullback-Leibler (KL) divergence between the predicted Dirichlet distribution and a non-informative prior. The Cross-Entropy Loss encourages the model to assign high probability to the true class:

$$\mathcal{L}_{\text{CE}}(x_i, y_i) = -\log(\frac{\alpha_{y_i}(x_i)}{S(x_i)}) = -\log(\frac{e_{y_i}(x_i) + 1}{\sum_{k=1}^{K}(e_k(x_i) + 1)}).$$

The KL divergence is given by:

$$\text{KL}(\text{Dir}(\boldsymbol{\alpha}) \| \text{Dir}(\mathbf{1})) = \log(\frac{B(\mathbf{1})}{B(\boldsymbol{\alpha})}) + \sum_{k=1}^{K}(\alpha_k - 1)(\psi(\alpha_k) - \psi(S)),$$

where $\psi(z) = \frac{d}{dz} \ln \Gamma(z)$ is the digamma function. The total loss for a single sample becomes:

$$\mathcal{L}(x_i, y_i) = -\log(\frac{e_{y_i}(x_i) + 1}{S(x_i)}) + \lambda \cdot \text{KL}(\text{Dir}(\boldsymbol{\alpha}(x_i)) \| \text{Dir}(\mathbf{1})).$$

### 4.2 UNCERTAINTY-GUIDED LAST LAYER RETRAINING

After training the evidential ERM model with the second-order risk, we leverage the estimated epistemic uncertainty $u(x)$ to empirically infer the posterior probabilities of the latent groups for each training sample. Specifically, we interpret $u(x)$ as an empirical prior over the group variable $g$, indicating the likelihood that a sample belongs to a majority or minority group. The uncertainty $u(x)$ is computed as:

$$u(x) = \frac{K}{S(x)} = \frac{K}{\sum_{k=1}^{K} \alpha_k(x)},$$

where $S(x)$ is the sum of the Dirichlet parameters from the evidential model. Higher values of $u(x)$ suggest larger epistemic uncertainty in the prediction for sample $x$.

We estimate the posterior group probabilities using these uncertainty-informed empirical priors:

$$\hat{p}(g \,|\, x, \theta) = u(x).$$

To perform Bayesian Model Averaging (BMA) during retraining, we integrate the estimated posterior group probabilities by reweighting the loss contributions of each sample based on $\hat{p}(g \,|\, x, \theta)$. The updated objective function for retraining the model parameters $\theta$ becomes:

$$\theta^* = \underset{\theta}{\arg\min} \sum_{i=1}^{N} u(x_i) \, \ell(f_\theta(x_i), y_i),$$

where $\ell(f_\theta(x_i), y_i)$ is the loss function for sample $i$. By reweighting the loss with the uncertainty estimates, we effectively average over models corresponding to different group assignments, mitigating the influence of group imbalances.

In practice, we only retrain the last layer of the model to learn the representative features like previous works Kirichenko et al. (2023); LaBonte et al. (2024b). This approach enhances the model's robustness to spurious correlations and improves generalization across all groups without requiring explicit group labels.

## 5 EXPERIMENTS

We evaluate *Learn from Known Unknowns* on a range of existing benchmarks and provide additional results to further support our motivation and methodology designs.

### 5.1 DATASETS

We first describe five datasets that we use in the experiments. These datasets are known for exhibiting poor worst-group performance due to spurious correlations, as highlighted in prior works. Our evaluation focuses on group robustness across both vision and language tasks: (1) **Colored MNIST** (Arjovsky et al., 2019) is a variant of the MNIST dataset for image classification where the digit images are colored to introduce spurious correlations between the color and the class label. (2) **Waterbirds** (Sagawa et al., 2020) is an image classification dataset with waterbird and landbird classes selected from the CUB dataset Welinder et al. (2010). The spurious attribute is the water or land backgrounds from the Places dataset Zhou et al. (2017), where more landbirds are present on land backgrounds than waterbirds, and vice versa. (3) **CelebA** (Liu et al., 2015) is an image classification with celebrity faces. The task is to identify whether the celebrity is blonde or not. The spurious attribute is gender where the majority group is blonde women in the training set. (4) **MultiNLI** (Williams et al., 2017) is a large-scale natural language inference dataset that contains sentence pairs across multiple genres, annotated with entailment labels (entailment, contradiction, or neutral). The spurious attribute involves syntactic or lexical patterns that correlate with specific labels. (5) **CivilComments** (Borkan et al., 2019) is a text classification dataset sourced from online comments. The task is to predict whether a comment is toxic or non-toxic. The spurious attributes include demographic identity terms (e.g., gender, race) that correlate with the label.

### 5.2 EXPERIMENTAL SETUPS

We begin by training all ERM models for both image and text datasets. For image datasets, we use a ResNet-50 (He et al., 2016) model pretrained on ImageNet as the backbone, while for text datasets, we employ the BERT-base-uncased model (Kenton & Toutanova, 2019) pretrained on the BookCorpus and English Wikipedia. We train the ERM models incorporating the Evidence Regularization to learn the uncertainty representations, as described in Section 4.1. Consistent with the approach in (Yang et al., 2023), we don't apply any data augmentations during ERM training. Model selection is based on the highest average accuracy on the validation set. For Uncertainty-Guided Retraining, we first estimate the uncertainty of the retraining samples using the ERM models, and then retrain only the last layer of the model to enhance computational efficiency. The retraining samples are randomly sampled from the misclassified portion of the training set and the validation set, which

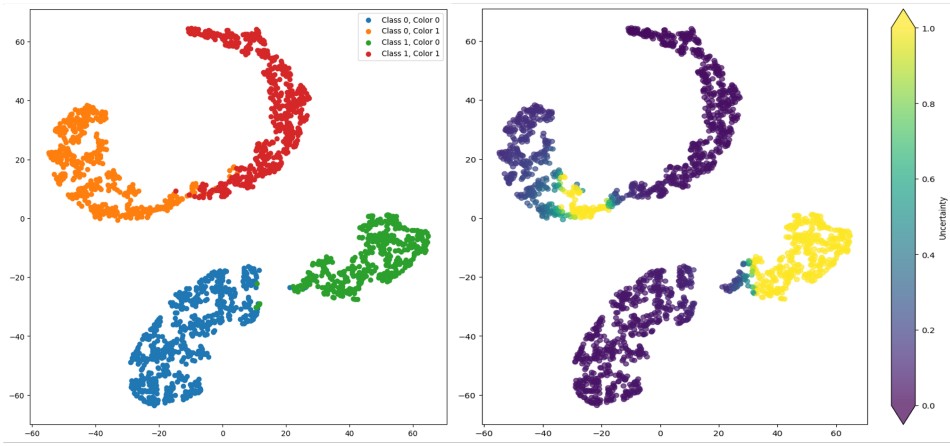

Figure 1: t-SNE visualization (Van der Maaten & Hinton, 2008) of the embeddings on the test set (left panel). The right panel shows the uncertainty quantification of these samples.

generally have higher uncertainty. This will prevent the model from overfitting the correctly classified data. The regularization coefficient $\lambda$ is dynamically set from 0 to 1 and the ratio of the current epoch number to a predefined annealing step, allowing $\lambda$ to smoothly increase from the initial value to 1 during training. Other hyperparameters, such as learning rates, are randomly sampled from predefined ranges at the start of training. We sample ten different hyperparameter configurations and select the best one based on validation performance. The final models are trained using three different random seeds with the selected hyperparameters, and we report the average accuracies along with the standard deviations. All experiments were conducted on NVIDIA A6000 GPUs. We add the hyperparameter and full experimental details in Appendix C.

### 5.3 SYNTHETIC EXPERIMENT

To demonstrate the concept of *Learn from Known Unknowns*, we first conducted a simple synthetic experiment using the Colored MNIST dataset (Arjovsky et al., 2019). In this experiment, we introduced spurious correlations between the color and the class label, with 90% of the training samples ($p_{\text{corr}} = 0.9$) exhibiting this color-class association, while only 10% of the test samples ($p_{\text{corr}} = 0.1$) retained the same correlation, as shown in Figure 2. For Class 0, there are 6,398 red instances and 344 green instances. For Class 1, there are 325 red instances and 5,526 green instances. Notably, the (0, green) and (1, red) represent the minority groups in this synthetic dataset.

We use *Learn from Known Unknowns* to learn group robustness models. First, we trained a LeNet-5 model (LeCun et al., 1998) using ERM with second-order risk minimization. After obtaining the uncertainty estimates from the ERM model, we retrained the last layer of the model with a reweighted loss to mitigate the impact of the spurious correlations.

We first show that the second-order risk minimization can give a good estimate of minority groups in Figure 1. As shown in Table 3, this approach significantly improved the performance for the minority group (Class 1, Color 0), with accuracy increasing from 3.74% to 84.58%. The performance of the majority groups remained unaffected by the retraining process. Here, the selection of digits 1 and 8 in this experiment is simply to provide a clear contrast between the two classes. Other digit pairs, such as 0 and 7, would yield similar results.

### 5.4 COMPARISON WITH BASELINE METHODS

In this subsection, we address three key questions: (1) How does *Learn from Known Unknowns* perform compared to methods that do not use group information during training? (2) What is the performance gap between *Learn from Known Unknowns* and oracle methods that leverage group annotations? (3) How efficient is our method compared to existing approaches?

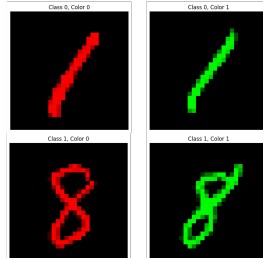

Table 2: Colored MNIST samples (two classes: digit 1 as class 0 and digit 8 as class 1) with spurious correlations by assigning the spurious attribute (color) $a$ to a portion $p_{\text{corr}}$ of the samples.

| Accuracy | ERM | Ours |
|---|---|---|
| Average | 76.34 | 96.21 |
| Class 0, Color 0 | 100.00 | 100.00 |
| Class 0, Color 1 | 82.81 | 97.83 |
| Class 1, Color 0 | 3.74 | 84.58 |
| Class 1, Color 1 | 100.00 | 99.45 |
| Worst Group | 3.74 | 84.58 |

Table 3: Comparison of Test Accuracy in each group between ERM and Ours (%).

**Comparison with Group-Label-Free Approaches:** To answer the first question, we compare our method against existing group-label-free approaches, including CVaR DRO (Levy et al., 2020), LfF (Nam et al., 2020a), JTT (Liu et al., 2021), and the recently proposed AFR (Qiu et al., 2023), as shown in Tables 4 and 5. *Learn from Known Unknowns* consistently achieves worst-group accuracy across three datasets, except for CelebA. This exception arises because CnC adopts a self-supervised learning paradigm, which is particularly effective for representation learning in CelebA. Notably, our approach employs a straightforward uncertainty quantification mechanism without introducing additional hyperparameters, which distinguishes it from other methods.

**Performance Gap with Oracle Methods:** Regarding the second question, we examine the performance gap between current group robustness methods and the oracle methods such as Group DRO (Sagawa et al., 2020), DFR (Kirichenko et al., 2023) and a recent method SELF (LaBonte et al., 2024b), which only require few annotations to infer group information. Our findings indicate that the performance gap is narrowing, suggesting that advancing group robustness requires a deeper understanding of the underlying problems rather than merely combining existing techniques.

**Computational and Memory Efficiency:** Our method introduces minimal computational overhead compared to existing approaches. It follows a two-phase process: training a standard ERM model with evidence regularization in the first phase, and retraining only the final layer in the second phase. In contrast, methods like JTT and CnC require retraining another new model from scratch. Thus, our approach is more efficient in terms of both computation and memory.

| Method | Backbone | Group Annotations | | Waterbirds | | CelebA | |
|---|---|---|---|---|---|---|---|
| | | Train | Val | Worst(%) | Average(%) | Worst(%) | Average(%) |
| ERM | ResNet50 | No | No | 72.6 | 97.3 | 47.2 | 95.6 |
| CVaR DRO (Levy et al., 2020) | ResNet50 | No | No | 75.5 | 89.9 | 60.2 | 95.1 |
| LfF (Nam et al., 2020a) | ResNet50 | No | No | 78.0 | 91.2 | 77.2 | 85.1 |
| JTT (Liu et al., 2021) | ResNet50 | No | No | 86.7 | 93.3 | 81.1 | 88.0 |
| CnC (Zhang et al., 2022) | ResNet50 | No | No | $88.5_{\pm0.3}$ | $90.9_{\pm0.1}$ | $\mathbf{88.8}_{\pm0.9}$ | $89.9_{\pm0.5}$ |
| AFR (Qiu et al., 2023) | ResNet50 | No | No | $90.4_{\pm1.1}$ | $94.2_{\pm1.2}$ | $82.0_{\pm0.5}$ | $91.3_{\pm0.3}$ |
| Ours | ResNet50 | No | No | $\mathbf{91.2}_{\pm0.6}$ | $95.3_{\pm0.3}$ | $84.3_{\pm2.3}$ | $91.6_{\pm0.9}$ |
| Group-DRO[†] (Sagawa et al., 2020) | ResNet50 | Yes | No | 91.4 | 93.5 | 88.9 | 92.9 |
| DFR[†] (Kirichenko et al., 2023) | ResNet50 | No | Yes | $92.9_{\pm0.9}$ | $94.2_{\pm0.3}$ | $88.3_{\pm1.1}$ | $91.3_{\pm0.5}$ |
| SELF (LaBonte et al., 2024b) | ResNet50 | No | Yes | $93.0_{\pm0.3}$ | $94.0_{\pm1.7}$ | $83.9_{\pm0.9}$ | $91.7_{\pm0.4}$ |

Table 4: Comparison to other group robustness methods in the image datasets, including standard deviations. [†] denotes oracle methods that use group annotations. The results of previous methods are from Nam et al. (2020b) and Yang et al. (2023).

## 5.5 CAN UNCERTAINTY CORRECTLY INFER GROUP INFORMATION?

A crucial aspect of *Learn from Known Unknowns* is determining whether the model's uncertainty quantification can effectively identify latent group information. In our experiments on the Waterbirds dataset, we used GradCAM visualizations (Figure 2) to examine the areas of focus for samples with

| Method | Backbone | Group Annotations | | MultiNLI | | CivilComments | |
|---|---|---|---|---|---|---|---|
| | | Train | Val | Worst(%) | Average(%) | Worst(%) | Average(%) |
| ERM | BERT | No | No | 67.9 | 82.4 | 57.4 | 92.6 |
| CVaR DRO (Levy et al., 2020) | BERT | No | No | 68.0 | 82.0 | 60.5 | 92.5 |
| LfF (Nam et al., 2020a) | BERT | No | No | 70.2 | 80.8 | 58.8 | 92.5 |
| JTT (Liu et al., 2021) | BERT | No | No | 72.6 | 78.6 | 69.3 | 91.1 |
| CnC (Zhang et al., 2022) | BERT | No | No | - | - | $68.9_{\pm2.1}$ | $81.7_{\pm0.5}$ |
| AFR (Qiu et al., 2023) | BERT | No | No | $73.4_{\pm0.6}$ | $81.4_{\pm0.2}$ | $68.7_{\pm0.6}$ | $89.8_{\pm0.6}$ |
| Ours | BERT | No | No | $\mathbf{74.5}_{\pm1.2}$ | $80.6_{\pm0.8}$ | $\mathbf{69.8}_{\pm1.6}$ | $92.2_{\pm0.8}$ |
| Group-DRO[†] (Sagawa et al., 2020) | BERT | Yes | No | 77.7 | 81.4 | 69.9 | 88.9 |
| DFR[†] (Kirichenko et al., 2023) | BERT | No | Yes | $63.8_{\pm0.8}$ | $80.2_{\pm0.6}$ | $64.4_{\pm1.1}$ | $80.7_{\pm0.2}$ |
| SELF (LaBonte et al., 2024b) | BERT | No | Yes | $70.7_{\pm2.5}$ | $81.2_{\pm0.7}$ | $79.1_{\pm2.1}$ | $87.7_{\pm0.6}$ |

Table 5: Comparison to other group robustness methods in the text datasets, including standard deviations. [†] denotes oracle methods that use group annotations. The results of previous methods are from Nam et al. (2020b) and Yang et al. (2023).

the highest and lowest uncertainty. We observed that high-uncertainty samples tended to highlight background regions associated with the spurious features, while low-uncertainty samples focused more on the birds themselves, representing the core features. Additionally, t-SNE embeddings (Figure 1) in our synthetic experiment revealed that high-uncertainty samples clustered around minority groups in the feature space, further supporting the correlation between uncertainty and group information. Quantitative analysis showed correlations between uncertainty values and true group labels across all datasets, indicating that uncertainty estimates are reliable proxies for identifying under-represented groups. These findings validate our approach of using uncertainty-based reweighting during retraining, as it allows the model to prioritize learning from challenging and minority group samples, thereby enhancing overall group robustness.

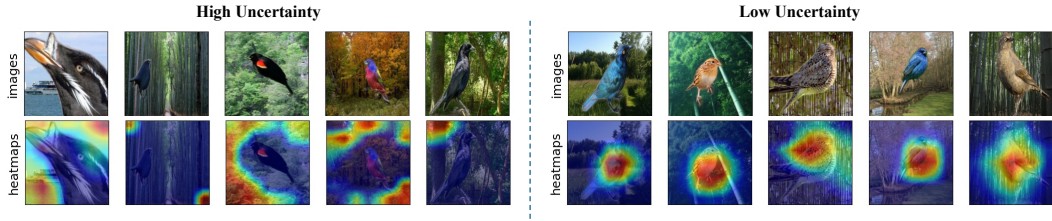

Figure 2: GradCAM (Selvaraju et al., 2020) visualizations for the top-5 images with the highest and lowest uncertainty in the Waterbirds dataset.

## 6 CONCLUSION

In this work, we introduce a new Empirical Bayesian perspective to enhance the understanding of group robustness. Our framework demonstrates that leveraging existing observations from both the ERM model and the dataset enables effective estimation of latent group information. Building on this, we propose an uncertainty-informed group robustness method, *Learn from Known Unknowns*, which utilizes epistemic uncertainty from biased ERM models to inform the retraining process. Extensive experiments demonstrate the efficacy of our approach, achieving compelling worst-group performance across diverse datasets. We believe this work provides valuable insights into developing more adaptive and scalable methods to improve group robustness in real-world applications.

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

# APPENDIX

This Appendix is organized as follows:

- Appendix A shows the pseudo code of *Learn from Known Unknowns*.

- Appendix B provides the data distribution on all datasets we experimented on.

- Appendix C provides additional experimental details.

- Appendix D shows the proof for the Theorem 3.1

## A   PSEUDO CODE OF LEARN FROM KNOWN UNKNOWNS

---

**Algorithm 1** *Learn from Known Unknowns*

---

1: **Input:** Dataset $D = \{(x_i, y_i)\}_{i=1}^N$, model $f_\theta$, number of epochs $T_1$, $T_2$
2: **Output:** Updated model parameters $\theta^*$
3: *Step 1: Evidential ERM Training*
4: **for** epoch $t = 1$ to $T_1$ **do**
5:     **for** each sample $(x_i, y_i) \in D$ **do**
6:         Estimate evidence $e_k(x_i)$ for each class $k$
7:         Compute Dirichlet parameters: $\alpha_k(x_i) = e_k(x_i) + 1$
8:         Compute expected class probabilities: $\mathbb{E}[p_k(x_i)] = \frac{\alpha_k(x_i)}{S(x_i)}$
9:         Compute classification loss: $\mathcal{L}_{\text{CE}}(x_i, y_i)$
10:        Compute KL Divergence: $\text{KL}(\text{Dir}(\boldsymbol{\alpha}(x_i)) \,\|\, \text{Dir}(\mathbf{1}))$
11:        Minimize total loss: $\mathcal{L}(x_i, y_i) = \mathcal{L}_{\text{CE}} + \lambda \cdot \text{KL}$
12:     **end for**
13: **end for**
14: *Step 2: Uncertainty-Guided Retraining*
15: **for** epoch $t = 1$ to $T_2$ **do**
16:     **for** each sample $(x_i, y_i) \in D$ **do**
17:         Estimate group probability: $\hat{p}(g \,|\, x_i, \theta) = u(x_i) = \frac{K}{S(x_i)}$
18:         Reweight loss with uncertainty: $u(x_i) \cdot \ell(f_\theta(x_i), y_i)$
19:     **end for**
20:     Update last layer parameters: $\theta^* = \underset{\theta}{\operatorname{argmin}} \sum_{i=1}^N u(x_i) \cdot \ell(f_\theta(x_i), y_i)$
21: **end for**
22: **Return:** Updated model parameters $\theta^*$

---

## B   DATASET DISTRIBUTIONS

By examining the distributions of the four datasets, we can observe distinct patterns of group/class imbalance. For the Waterbirds and MultiNLI datasets, although group imbalances are present, the overall class distribution is nearly balanced. In contrast, both group and class imbalances are observed in the CelebA and CivilComments datasets. This variation suggests that applying simple techniques, such as class-balancing or group-balancing, may have different impacts on improving group robustness, depending on the specific characteristics of the dataset. Recent work by LaBonte et al. (2024a) also emphasizes that existing class-balancing strategies can produce inconsistent results across these datasets. This underscores the need for a deeper understanding of group robustness, particularly in real-world applications. Consequently, it is crucial to develop adaptive methods that can more effectively address both class and group imbalances to enhance group robustness across diverse datasets.

| Dataset | Class $y$ | Spurious $a$ | Train | Val | Test |
|---|---|---|---|---|---|
| *Waterbirds* | | | | | |
| Waterbirds | landbird | land | 3498 | 467 | 2225 |
| Waterbirds | landbird | water | 184 | 466 | 2225 |
| Waterbirds | waterbird | land | 56 | 133 | 642 |
| Waterbirds | waterbird | water | 1057 | 133 | 642 |
| *CelebA* | | | | | |
| CelebA | non-blond | female | 71629 | 8535 | 9767 |
| CelebA | non-blond | male | 66874 | 8276 | 7535 |
| CelebA | blond | female | 22880 | 2874 | 2480 |
| CelebA | blond | male | 1387 | 182 | 180 |
| *MultiNLI* | | | | | |
| MultiNLI | contradiction | no negation | 57498 | 22814 | 34597 |
| MultiNLI | contradiction | negation | 11158 | 4634 | 6655 |
| MultiNLI | entailment | no negation | 67376 | 26949 | 40496 |
| MultiNLI | entailment | negation | 1521 | 613 | 886 |
| MultiNLI | neither | no negation | 66630 | 26655 | 39930 |
| MultiNLI | neither | negation | 1992 | 797 | 1148 |
| *CivilComments* | | | | | |
| CivilComments | neutral | no identity | 148186 | 25159 | 74780 |
| CivilComments | neutral | identity | 90337 | 14966 | 43778 |
| CivilComments | toxic | no identity | 12731 | 2111 | 6455 |
| CivilComments | toxic | identity | 17784 | 2944 | 8769 |

Table 6: Data quantities for each group in the datasets. The detail numbers are credited to LaBonte et al. (2024b)

## C  MORE EXPERIMENTAL DETAILS

| Hyperparameters | Waterbirds | CelebA | MultiNLI | CivilComments |
|---|---|---|---|---|
| initial learning rate | 3e-3 | 3e-3 | 1e-5 | 1e-3 |
| number of epochs | 100 | 20 | 10 | 10 |
| learning rate scheduler | CosineAnnealing | CosineAnnealing | Linear | Linear |
| optimizer | SGD | SGD | AdamW | AdamW |
| weight decay | 1e-4 | 1e-4 | 1e-4 | 1e-4 |
| batch size | 32 | 128 | 16 | 16 |
| backbone | ResNet-50 | ResNet-50 | BERT-base | BERT-base |

Table 7: Hyperparameters for evidential ERM training.

| Hyperparameters | Waterbirds | CelebA | MultiNLI | CivilComments |
|---|---|---|---|---|
| learning rate | 1e-3 | 1e-3 | 1e-5 | 1e-3 |
| number of epochs | 100 | 100 | 300 | 300 |
| number of batches / epoch | 200 | 200 | 300 | 300 |
| optimizer | SGD | SGD | SGD | SGD |
| batch size | 128 | 128 | 128 | 128 |

Table 8: Hyperparameters used in Uncertainty-guided Retraining.

## D    PROOF OF THEOREM 3.1

*Proof.* We aim to show that under the given assumptions, the posterior mean of the group variable $g$ given $x$, $y$, and $\theta$ satisfies:

$$\mathbb{E}[g|x, y, \theta] = \mathbb{E}[g] + \sigma^2 \frac{\partial}{\partial y} \log p(y|x, \theta).$$

Starting with the definition of posterior expectation:

$$\mathbb{E}[g|x, y, \theta] = \sum_{g \in \mathcal{G}} g \cdot p(g|x, y, \theta).$$

Using Bayes' theorem, the posterior probability $p(g|x, y, \theta)$ is:

$$p(g|x, y, \theta) = \frac{p(g) \cdot p(y|x, g, \theta)}{p(y|x, \theta)}.$$

Substituting this into the expectation:

$$\mathbb{E}[g|x, y, \theta] = \frac{1}{p(y|x, \theta)} \sum_{g \in \mathcal{G}} g \cdot p(g) \cdot p(y|x, g, \theta).$$

Next, consider the log-marginal likelihood:

$$\log p(y|x, \theta) = \log \left( \sum_{g \in \mathcal{G}} p(g) \cdot p(y|x, g, \theta) \right).$$

Differentiating both sides with respect to $y$:

$$\frac{\partial}{\partial y} \log p(y|x, \theta) = \frac{1}{p(y|x, \theta)} \sum_{g \in \mathcal{G}} p(g) \cdot \frac{\partial p(y|x, g, \theta)}{\partial y}.$$

Given that $p(y|x, g, \theta)$ belongs to the exponential family, it can be expressed as:

$$p(y|x, g, \theta) = h(y) \exp \left( \eta(g, \theta)^\top T(y) - A(g, \theta) \right),$$

where $\eta(g, \theta)$ is the natural parameter vector, $T(y)$ is the sufficient statistic vector, $A(g, \theta)$ is the log-partition function ensuring normalization.

Differentiating $p(y|x, g, \theta)$ with respect to $y$:

$$\frac{\partial p(y|x, g, \theta)}{\partial y} = p(y|x, g, \theta) \cdot \left( \frac{\partial \log h(y)}{\partial y} + \eta(g, \theta)^\top \frac{\partial T(y)}{\partial y} \right).$$

Assuming $\frac{\partial \log h(y)}{\partial y} = 0$ for simplicity (i.e., $h(y)$ is constant or its derivative does not depend on $g$):

$$\frac{\partial p(y|x, g, \theta)}{\partial y} = p(y|x, g, \theta) \cdot \eta(g, \theta)^\top \frac{\partial T(y)}{\partial y}.$$

Substituting this into the derivative of the log-marginal likelihood:

$$\frac{\partial}{\partial y} \log p(y|x, \theta) = \frac{1}{p(y|x, \theta)} \sum_{g \in \mathcal{G}} p(g) \cdot p(y|x, g, \theta) \cdot \eta(g, \theta)^\top \frac{\partial T(y)}{\partial y}.$$

Simplifying, we get:

$$\frac{\partial}{\partial y} \log p(y|x, \theta) = \sum_{g \in \mathcal{G}} p(g|x, y, \theta) \cdot \eta(g, \theta)^\top \frac{\partial T(y)}{\partial y}.$$

Assuming a linear relationship between the natural parameter and the group variable, i.e., $\eta(g, \theta) = \theta \cdot g$, we substitute:

$$\frac{\partial}{\partial y} \log p(y|x, \theta) = \sum_{g \in \mathcal{G}} p(g|x, y, \theta) \cdot (\theta \cdot g)^\top \frac{\partial T(y)}{\partial y} = \theta^\top \left( \sum_{g \in \mathcal{G}} g \cdot p(g|x, y, \theta) \right) \frac{\partial T(y)}{\partial y}.$$

Rearranging the expression:

$$\mathbb{E}[g|x, y, \theta] = \frac{1}{\theta^\top \frac{\partial T(y)}{\partial y}} \cdot \frac{\partial}{\partial y} \log p(y|x, \theta).$$

Finally, introducing the prior mean $\mathbb{E}[g]$ and the variance $\sigma^2$ of the sufficient statistics $T(y)$, we approximate:

$$\mathbb{E}[g|x, y, \theta] \approx \mathbb{E}[g] + \sigma^2 \cdot \frac{\partial}{\partial y} \log p(y|x, \theta).$$

Here, $\sigma^2$ represents the variance of $T(y)$ under $p(y|x, g, \theta)$, serving as a scaling factor analogous to the variance in the original Tweedie's formula.

$\square$

