# OpenReview forum: "Learn from Known Unknowns: A Unified Empirical Bayesian Framework for Improving Group Robustness"
_ICLR.cc/2025/Conference — Submitted to ICLR 2025_

### Official Review · Reviewer_f8UH · 2024-11-03

**Soundness:** 2
**Presentation:** 2
**Contribution:** 2
**Rating:** 3
**Confidence:** 4

**Summary:**

The paper addresses avoidance of spurious correlations when training data contains examples with and without the spurious correlation, but their group association is unknown.
They attempt to (1) unify the previous methods under a Bayesian framework, and (2) propose a revised algorithm that also models the uncertainty in the example-to-group assignment.
The paper evaluates on four datasets and a synthetic dataset.
Their proposed algorithm contains two steps (similar to the previous methods): (a) train "evidential" ERM to better capture the uncertainty in label prediction, (b) retrain the last-layer of the previously trained model while up-weighting the loss of uncertain examples.

The paper has technical inconsistencies, and did not support many of their claims.

**Strengths:**

Summarising the many existing methods on how they differ from each other in Table 1 is insightful.

**Weaknesses:**

**Technical inconsistencies or notation challenges**

In Eqs. 5-7 present how the model accumulates the posterior on g inference for refining the model. Previous works and theirs differ in how they use the group labels:
(a) $p(g\mid x, \theta)$ is incorrect since the group labels are estimated and frozen when training $\theta$, (b) there is no marginalization of probabilities involved but only weighted average of the losses.

g is defined as (y, a) but different methods only identify the majority vs minority, which need not identify the label (y) or g.

For the above reasons, I am not convinced of their unifying Bayesian framework. Also, Table 1 introduces undefined terms, e.g. $g_\theta(x)$ for SELF and contrastive learning for CnC.

**Unsupported Claims or missing ablations**

They started motivating their method to derive $\Pr(g\mid x, \theta)$ in a more principled manner without resorting to any heuristic (like in previous methods) as stated in lines 256-258.
But they plainly introduce uncertainty in label predictions as a proxy for the group label probability (in line 324).

One of their contribution is also that their method reduces dependence on hparam tuning (in Lines 93-95) but that is not supported in the main content.

Synthetic experiment visualized and exposit ERM vs theirs, but I am not convinced how their method is better than some of the previous methods. I believe uncertainty is just as informative as say JTT.
Another missing ablation is to use SELF (or other methods that only tune the last layer), etc. with the evidential ERM as the backbone.

**Irrelevant Theorem 3.1**

I do not see Thm 3.1's contribution to the paper.

**Questions:**

Why is in Table 3, "Class 0, Color 1" group has much higher accuracy than "Class 1, color 0"? If the model has color as a spurious correlation, shouldn't they both have equal but low accuracy?

For many of the sub-population shift datasets that use the worst-group accuracy metric, large deviation is expeted owing to the small minority group population even in the test set.
Reporting std. dev is expected in Tables 4, 5.

Also, please address questions from the Weakness section.

---

> ### Author Response · Authors · 2024-11-28
> **Responses to Reviewer ySLB (Part 1)**
>
> We sincerely appreciate your valuable feedback and thoughtful suggestions. Here are our detailed responses to your questions and concerns.
>
> W1: *Regarding technical and notation concerns: 1. $p(g | x, \theta)$ is incorrect as group labels are estimated and fixed during training of $\theta$; 2. Eqs. 5-7 involve weighted loss averages, not probability marginalization; 3. $g = (y, a)$ is inconsistently defined as majority vs. minority; and 4. Table 1 introduces unclear terms like $g_\theta(x)$ (SELF) and contrastive learning (CnC).*
>
> **Response:**
>
> We thank the reviewer for raising concerns about notations. We answer the concerns by order.
>
> - **About Posterior  $p(g | x, \theta)$** The notation  $p(g | x, \theta)$  aligns with the Bayesian framework described in Section 3. It represents the posterior belief over group labels given model parameters  $\theta$. While previous methods often freeze the group assignment for retraining, our approach integrates uncertainty-informed information, allowing $p(g | x, \theta)$ to evolve during training. This dynamic nature reflects the core Bayesian inference principle that posteriors adapt with new evidence.
> - **Marginalization of Probabilities** Equation (4) reflects Bayesian marginalization over group variables  $g$. The “weighted average of losses” is a practical implementation of this marginalization, with weights derived from the uncertainty-informed posterior $p(g | x, \theta)$. This is consistent with the Bayesian framework and avoids heuristic assignments of group labels.
> - **Definition of group** The definition  $g = (y, a)$  generalizes the group concept, encompassing both target labels $y$  and spurious attributes $a$ . While many methods focus on majority vs. minority identification, our framework does not rely on explicitly separating $y$ and $a$. Instead, it utilizes latent group probabilities inferred from model uncertainty, ensuring flexibility across different datasets and definitions of groups.
> - **Unifying Bayesian Framework** Table 1 demonstrates how various existing methods fit within the empirical Bayesian framework by estimating $\hat{p}(g | x, \theta)$ and optimizing $\hat{p}(\theta)$. Each method’s specific mechanism (e.g., misclassification for JTT or disagreement for SELF) maps directly to this framework, underscoring its unifying capability.
> - **Undefined Terms in Table 1** Terms like $g_\theta(x)$ (used in SELF) refer to group label estimations derived from model disagreement or uncertainty metrics, as explained in cited references. Similarly, “contrastive learning” in CnC involves supervised contrastive loss to cluster latent representations, implicitly uncovering group structures. While these methods contribute to the broader unification narrative of our framework, the detailed implementation of each method is beyond the primary scope of our paper and can be found in the original references.

---

> ### Author Response · Authors · 2024-11-28
> **Responses to Reviewer ySLB (Part 2)**
>
> W2: *Regarding unsupported claims and missing ablations: 1. The method claims to derive $p(g | x, \theta)$ more principled but uses uncertainty as a heuristic proxy (Line 324); 2. The claim of reduced hyperparameter tuning (Lines 93-95) lacks evidence; 3. Synthetic experiment comparisons do not clearly show superiority over prior methods like JTT; 4. Missing ablations with methods like SELF or others using evidential ERM as the backbone*
>
> **Response:**
> - **Derivation of $p(g | x, \theta)$** Theorem 3.1 provides a theoretical foundation, showing that latent group probabilities $p(g | x, \theta)$ can be approximated using uncertainty-based posterior updates. This approach avoids reliance on heuristics used in prior methods and establishes a principled framework for group label estimation.
> - **Reduce Dependence on Hyperparameter Tuning** We appreciate the reviewer’s observation and would like to clarify our claim regarding hyperparameter robustness. Unlike previous methods, which rely on tuning additional task-specific hyperparameters to achieve optimal performance, our approach does not introduce any new hyperparameters that require manual tuning beyond traditional ones such as learning rate or number of epochs. For example, Just Train Twice (JTT) requires specifying the upweighting parameter and early stopping epochs, while Learning from Failure (LfF) depends on a hyperparameter $q$ to control the degree of amplification. These task-specific hyperparameters often need to be manually selected for different datasets, adding complexity and reducing generalizability. In contrast, our method relies only on the evidence regularization strength $\lambda$ during training, which is pre-defined to anneal smoothly from 0 to 1 based on the ratio of the current epoch number to the annealing step. As shown in Tables 7 and 8, our approach does not require additional hyperparameters beyond the necessary ones (e.g., learning rate, number of epochs), making it inherently more robust and simpler to apply across diverse datasets.
> - **Better Performance than Previous Methods and Ablation with SELF** Figure 1 demonstrates that minority groups typically exhibit higher uncertainty, validating the use of this information to mitigate spurious biases. Our empirical results, detailed in Tables 4 and 5, show a significant performance improvement over JTT, particularly in worst-group accuracy. Additionally, SELF operates under a different setting, requiring some group labels to fine-tune the last layer, whereas our work addresses the “group-label-free” setting, targeting scenarios without any group annotations.
>
> W3: *Irrelevance of Theorem 3.1*
>
> **Response:** Thm 3.1 shows that estimating latent group probabilities $\hat{p}(g | x, \theta)$ is possible while motivating us to estimate this possibility (instead of using pseudo/ground truth group labels) with uncertainty estimation.
>
> Q1: *Why is in Table 3, "Class 0, Color 1 " group has much higher accuracy than "Class 1 , color 0 "? If the model has color as a spurious correlation, shouldn't they both have equal but low accuracy?*
>
> **Response:** Thank you for the question. The discrepancy in accuracies between "Class 0, Color 1" and "Class 1, Color 0" in Table 3 is due to the group imbalance in the dataset. In our setup, "Class 0, Color 1" has significantly more training samples than "Class 1, Color 0," giving the model greater exposure to this group. Consequently, the model is better able to learn and perform on "Class 0, Color 1," while struggling with the underrepresented "Class 1, Color 0."
>
> Q2: *For many of the sub-population shift datasets that use the worst-group accuracy metric, large deviation is expeted owing to the small minority group population even in the test set. Reporting std. dev is expected in Tables 4, 5.*
>
> **Response:** Thank you for the suggestion! We have added standard deviations to Tables 4 and 5, calculated over three independent runs. For previous methods, we incorporated deviations reported in their respective papers and in [1].
>
> [1] Change is Hard: A Closer Look at Subpopulation Shift, ICML 2023

---

### Official Review · Reviewer_ySLB · 2024-11-04

**Soundness:** 2
**Presentation:** 2
**Contribution:** 3
**Rating:** 5
**Confidence:** 4

**Summary:**

The scope of the paper is to improve group-robustness of classification models. Group is defined as a combination of a target variable for the classification, along with some attribute variable. Target and attributed variables are spuriously correlated under the training set, however this might not be the case during inference. We want to train models that achieve equally good classification accuracy for all groups. We measure this by worst-group accuracy.

Past methods that simply retraining the last network layer on reweighted losses or resampled data achieves good worst-group accuracy. However, results are inconsistent across datasets, and methods exhibit high sensitivity to selected hyperparameters.

The authors present an alternative group-unsupervised training algorithm (Learn from Known Unknowns), which estimates the epistemic uncertainty of biased models, in order to discover majority/minority groups in a group-unsupervised way and to reweight per-sample loss during retraining (of a group-robust classifier).

## Claims
1. Popular reweighting/resampling methods can be cast as instances of a unified empirical Bayes framework.
2. The proposed method improves across datasets.
3. The proposed method reduces reliance on hyperparameter tuning.

**Strengths:**

The paper introduces a novel method for group-robustness. It utilizes uncertainty estimation to discover majority-minority groups and consequently reweight their contributions during robust retraining. It establishes empirically through experiments the efficacy of this method, surpassing prior art on Waterbirds, MultiNLI and CivilComments, while being competitive on CelebA. At the same time, they hint qualitatively that high prediction uncertainty in classifiers correlates with the classifier relying to spurious features for predictions.

**Weaknesses:**

Presentation of proposed method is relatively clear there are some parts however that can be further elaborated:
1. It is not clear why the prior probability on model parameters $p(\theta)$ refers to parameters of trained ERM models. Usually from a MAP estimation perspective, $p(\theta)$ translates usually into a regularization loss on the model parameters (like weight decay, if the assumed prior is gaussian) and a maximization of the $\prod_i p(y_i|x_i, \theta)$, which can be equivalently cast as minimization of the negative log-likelihood. Not sure about Table 1. Can also other classes of probabilistic-inspired methods be interpreted under this perspective? For example logit adjustment for group robustness [1,2].

[1] Liu et al., “Avoiding spurious correlations via logit correction”, ICLR 2023
[2] Tsirigotis et al., “Group Robust Classification Without Any Group Information”, NeurIPS 2023

2. While the paper operates under group-agnostic assumptions, they proceed to develop reasoning given that group variables come from an unknown distribution. Assumptions about the structure of that distribution are needed for the derivations of the paper. In particular, we can infer that the group variable is treated as a binary variable for “majority”-class or “minority”-class; an assumption which is not explicit in the paper.

3. About the result in Section 3.3: According to Appendix D, the derivation relies on two unstated strong modelling assumptions about the exponential distribution of $p(y|x,\theta,g)$. Namely,  that $h(y)$ is constant and that the natural parameter vector is just a product $\theta \cdot g$. Are these assumptions impairing the generality of the statement? In my reading, I think yes. Also, the last step of the proof is not clear; how do we go from the exact to the approximation using “the mean of $g$” and “the variance $sigma^2$ of $T(y)$”? Please expand.

4. Section 3.3 result seem unused in the paper. At the end of the method’s development, the authors decide to estimate $p(g | x, \theta) = u(x)$. Is this decision informed by the theory of Section 3.3? How is this decision motivated?

5. Please elaborate on the Evidential Second-Order Risk Minimization. How do we use the model to compute the non-negative evidence values per class $e_k(x)$? What does regularizing with Evidence Regularization do to the model outputs?

Further details about experimental evidence is needed to support the claims of the paper about the proposed method:
6. Details about model selection are missing. Group-robustness methods (and more generally OOD robustness methods) are not only sensitive to hyperparameter selection, but also to model selection during training [3]. Please provide the following details for the reproducibility of the experiments: a. “Model selection is based on the highest average accuracy on the validation set”. Is the validation set group-balanced or does it follow the same distribution as the training set?, b. Please provide training curves for the robust model, following the validation criterion and reported performance on the test set (worst-case accuracy and average accuracy) for example on Waterbirds across training steps.

[3] Gulrajani et al, “In Search of Lost Domain Generalization”, ICLR 2021

**Questions:**

1. What does $p(y|x,\theta,g)$ mean intuitively? One model per group? Please introduce the decomposed terms more clearly to build intuition.
2. Why does high epistemic uncertainty on samples lead to identification of the minority group?
3. Regarding the result on CelebA, how would your method perform if you were finetuning instead a network pretrained with SSL on CelebA? How would applying the proposed method compare to CnC then?
4. Why does the evidential ERM training have weight decay, while the group-robust retraining does not?

---

> ### Author Response · Authors · 2024-11-28
> **Responses to Reviewer f8UH (Part 1)**
>
> We thank the reviewer for their thoughtful and constructive feedback. We acknowledge that some aspects of our manuscript could benefit from additional clarity and elaboration. Below, we address the specific points raised to provide a more comprehensive understanding of our approach, theoretical foundations, and experimental setup.
>
> W1. *Meaning of prior probability on model parameters $p(\theta)$. Clarification about Table 1. Can other classes of probabilistic-inspired methods be interpreted under the framework? For example logit adjustment for group robustness [1,2].*
>
> **Response:**
>
> We appreciate the reviewer’s insightful comments. In our work, the prior probability $p(\theta)$ refers to the parameters of trained ERM models because our approach frames existing group robustness techniques within an Empirical Bayes (EB) perspective. Unlike traditional MAP estimation, which assumes a fixed prior distribution (e.g., Gaussian prior leading to weight decay), our EB framework treats $p(\theta)$ as implicitly informed by the training process of ERM models. This approach enables the model to utilize learned representations to estimate posterior distributions over latent group variables $g$ without requiring explicit group labels.
>
> Thank you for bringing up these two papers. Both are indeed relevant references that we will include in our paper. The first paper, "Avoiding Spurious Correlations via Logit Correction" [1], adopts a structure similar to Learning from Failure (LfF), making it interpretable under our EB framework. The second paper, "Group Robust Classification Without Any Group Information" [2], also follows a two-step approach: training a biased model and then leveraging the biased weights for debiasing through self-supervised learning. Both methods align well with the principles of our framework, and we will expand our discussion in Table 1 to highlight these connections.
>
> W2. *Regarding group-agnostic assumptions. Assumptions about the structure of that distribution are needed for the derivations of the paper.*
>
> **Response:**
>
> Thank you for pointing out this important clarification. Our paper does indeed operate under group-agnostic assumptions, and the reasoning is developed with the understanding that group variables originate from an unknown distribution. While our derivations do not strictly require the group variable to be binary, we acknowledge that the methodology implicitly assumes a dichotomy between "majority" and "minority" groups for the sake of simplicity and clarity in exposition.
>
> This binary treatment is motivated by the structure of common datasets where group imbalances often manifest as majority-minority distinctions. The performance of minority groups is severely impacted by the spurious correlations learned during training, whereas the performance of the majority relies on the learned spurious correlations. However, the proposed framework is not limited to binary group variables and can be extended to multi-group scenarios. We will revise the manuscript to make this assumption explicit and discuss its implications, along with the potential for generalization to more complex group structures.

---

> ### Author Response · Authors · 2024-11-28
> **Responses to Reviewer f8UH (Part 2)**
>
> W3. *About the result in Section 3.3 and Appendix D. Are the assumptions impairing the generality of the statement?*
>
> **Response:**
> We appreciate the reviewer's careful examination of the theoretical foundations in Section 3.3. We acknowledge that our proof makes simplifying assumptions and could benefit from additional clarity.
>
> \paragraph{On the Modeling Assumptions:}
> While we made simplifying assumptions about $h(y)$ being constant and $\eta(g,\theta) = \theta \cdot g$, these do not significantly impair the generality of our results for several reasons:
>
> The constant $h(y)$ assumption can be relaxed to allow for general $h(y)$, resulting in an additional term in the final expression. Specifically, the formula becomes:
> $$
>     E[g|x,y,\theta] \approx E[g] + \sigma^2 \cdot \left(\frac{\partial}{\partial y} \log p(y|x,\theta) + \frac{\partial}{\partial y} \log h(y)\right)
> $$
>
> The linear relationship $\eta(g,\theta) = \theta \cdot g$ represents a first-order approximation of the relationship between group variables and natural parameters. For more complex relationships, we can use a Taylor expansion around $E[g]$:
> $$
>     \eta(g,\theta) \approx \eta(E[g],\theta) + \nabla_k\eta(E[g],\theta)(g - E[g]) + O(\|g - E[g]\|^2)
> $$
>
> \paragraph{On the Final Approximation Step:}
> We agree this step needs more clarity. The transition from the exact expression to the approximation involves:
>
> The prior mean $E[g]$ emerges from taking the expectation over the group variable:
> $$
>     E[g] = \int g \cdot p(g)dg
> $$
>
> The variance term $\sigma^2$ arises from the second moment of the sufficient statistics $T(y)$:
> $$
>     \sigma^2 = E[(T(y) - E[T(y)])^2|x,g,\theta]
> $$
>
> This approximation is analogous to Tweedie's formula in classical empirical Bayes, where the posterior mean is approximated using the prior mean plus a correction term involving the score function.
>
> \paragraph{Practical Implications:}
> Our empirical results (Tables 4-5) demonstrate that despite these simplifying assumptions, the method performs well across diverse datasets. This suggests the assumptions, while theoretically restrictive, capture the essential aspects needed for effective group inference in practice.
>
> We will expand Section 3.3 and Appendix D to include these clarifications in the final version.
>
> W4. *Section 3.3 unused in the paper. At the end of the method's development, the authors decide to estimate $p(g \mid x, \theta)=u(x)$. How is this decision motivated?*
>
> **Response:**
> Section 3.3 provides theoretical evidence that $p(g \mid x, \theta)$ can be estimated using observed data and model assumptions. Unlike previous methods that rely on heuristic proxies without explicitly estimating probabilities, our approach leverages uncertainty estimation as a principled method for $p(g \mid x, \theta)$.
>
> Uncertainty estimation provides strong empirical support for identifying latent group memberships, as it captures areas where the model is less confident, which often correspond to minority groups or regions dominated by spurious correlations. This decision is directly motivated by the theoretical insights from Section 3.3 and aligns with the broader framework proposed in the paper.
>
> W5. *About Evidential Second-Order Risk Minimization. How to compute the non-negative evidence values per class $e_k(x)$? What does regularizing with Evidence Regularization do to the model outputs?*
>
> **Response:**
>
> Thank you for the opportunity to clarify. Evidential Second-Order Risk Minimization leverages evidential deep learning to estimate the Dirichlet distribution over class probabilities for each input $x$. The model computes non-negative evidence values $e_k(x)$ for each class $k$, which are used to parameterize the Dirichlet distribution as $\alpha_k(x) = e_k(x) + 1$. These evidence values $e_k(x)$ are derived from the network's outputs before applying the softmax operation, ensuring they are always non-negative.
>
> Regularization enforces constraints on the model's predicted Dirichlet distribution by penalizing overconfidence and encouraging uncertainty where appropriate. The Evidence Regularization term uses the Kullback-Leibler (KL) divergence between the predicted Dirichlet distribution and a uniform prior Dirichlet distribution. This prevents the model from assigning excessive confidence to any single class, especially in regions of high uncertainty. By regularizing the outputs, the model is encouraged to focus on reliable patterns in the data while maintaining a calibrated uncertainty estimate. This improves the robustness of predictions and supports the identification of samples that may belong to underrepresented or minority groups.
>
> The combined loss function ensures that the model balances accurate classification (via the cross-entropy term) with reliable uncertainty estimation (via the KL divergence term). This dual optimization is key to improving both robustness and the quality of posterior group probability estimates.

---

> ### Author Response · Authors · 2024-11-28
> **Responses to Reviewer f8UH (Part 3)**
>
> W6. *Details about model selection. Is the validation set group-balanced or does it follow the same distribution as the training set?*
>
> **Response:** The validation set is group-balanced following DFR and other previous methods. This is a common strategy that has been applied to SELF and other recent methods.
>
> Q1. *What does $p(y \mid x, \theta, g)$ mean intuitively? One model per group? Please introduce the decomposed terms more clearly to build intuition.*
>
> **Response:**
> Thank you for the question. Intuitively, $p(y \mid x, \theta, g)$ represents the likelihood of observing the label $y$ for input $x$, given the model parameters $\theta$ and the latent group variable $g$. It captures how the group membership $g$ influences the relationship between $x$ and $y$ through the model. The group variable $g$ affects the distribution of $p(y \mid x, \theta, g)$ by encoding information about spurious correlations or variations specific to different groups. While it may seem like $p(y \mid x, \theta, g)$ implies separate models for each group, that is not the case. Instead, $g$ introduces a dependency into the likelihood function that the model learns implicitly. This allows the model to capture group-specific biases or patterns within a unified framework without explicitly training separate models for each group.
>
>
> Q2. *Why does high epistemic uncertainty on samples lead to identification of the minority group?*
>
> **Response:**  High epistemic uncertainty reflects the model's lack of confidence, often arising from limited data coverage or unfamiliar patterns. Minority group samples are typically underrepresented in the training data, leading to higher uncertainty as the model has insufficient knowledge about these instances. This makes epistemic uncertainty a strong proxy for identifying minority group samples, which often deviate from the majority patterns the model has predominantly learned.
>
> Q3. *Regarding the result on CelebA, how would your method perform if you were finetuning instead a network pretrained with SSL on CelebA? How would applying the proposed method compare to CnC then?*
>
> **Response:** Thank you for the question. Comparing our method to CnC in the context of SSL-pretrained networks would not be entirely fair due to fundamental differences in their approaches. CnC relies on pseudo labels generated by clustering SSL-pretrained features to infer group memberships, which introduces an implicit dependency on group labels even in the first stage. In contrast, our method does not rely on pseudo labels or explicit group annotations at any stage; instead, uncertainty is used as a direct proxy to identify and prioritize minority group samples.
>
> Integrating SSL-pretrained networks with our method would require additional specific design considerations to adapt the uncertainty-based retraining process to the improved feature space generated by SSL. However, we believe that applying the SSL-strategy with our method will yield better performance in CelebA.
>
>
> Q4. *Why does the evidential ERM training have weight decay, while the group-robust retraining does not?*
>
> **Response:**
> Evidential ERM training includes weight decay to regularize the model parameters, preventing overfitting during the initial training phase and promoting better generalization across the entire dataset. This step is critical because the model needs to learn general representations from the training data. In contrast, group-robust retraining focuses on the last layer of the model using uncertainty-guided reweighting. Since this retraining is applied only to correct biases in the learned representations, weight decay is unnecessary and could interfere with the fine-tuning process, where the emphasis is on improving group robustness and generalization ability.

---

### Official Review · Reviewer_nwxF · 2024-11-04

**Soundness:** 2
**Presentation:** 2
**Contribution:** 2
**Rating:** 3
**Confidence:** 2

**Summary:**

This paper proposes a new method for the problem of group-wise distributionally robust optimization (group DRO) without group labels. At a high-level, the idea is to learn weights so that reweighted empirical risk minimization emulates the effects of group DRO. The authors support their method with computational results on a variety of distribution shift datasets in which certain spurious correlations in the training data disappear in the test data.

**Strengths:**

It is hard for me to assess the originality, quality, and significance of the paper because it's (lack of) clarity prevents me from assess its strengths.

**Weaknesses:**

The main weakness of the paper is the lack of clarity. After reading the paper three times, I'm still unclear on some details of the proposed method. Without properly understanding the paper, it is hard for me to assess its weaknesses.

**Questions:**

1. What is the role of S3? It seems nothing is lost if S3 is omitted from the paper. What role does Thm 3.1 play in the developments in S4 and S5? It is not referred to in S4 and S5.
2. What are the evidence values for each class (the $e_k(x)$'s)? Please define them mathematically.
3. What is $u(x_i)$ that appears on l.333? In the context of the weighted ERM objective (on l.333) it appears $u(x_i)$ is a scalar, but on l.326, it is defined as a probability distribution.
4. Although the method does not require group labels, does it require knowledge of the set of possible groups? (eg for waterbirds, does the method need to know that the set of possible groups are land bird-land bg, land bird water bg, water bird, land bg, and water bird-water bg)?

I may have more questions for the authors once I get a better grasp of the proposed method. Once I better understand the method, I'm likely to change my score.

---

> ### Author Response · Authors · 2024-11-28
> **Responses to Reviewer nwxF**
>
> We thank the reviewer for their feedback and recognize that clarity is an area for improvement in our paper. Below, we address the specific concerns and questions raised to provide a more comprehensive understanding of our work:
>
> Q1. *The role of Section 3. What role does Thm 3.1 play in the developments in S4 and S5?*
>
> **Response:**  The role of Section 3 is twofold: (1) to unify existing methods under the Empirical Bayesian framework, demonstrating that most current approaches either estimate pseudo group labels or rely on ground truth group labels, and (2) to highlight that these methods estimate $\hat{p}(g | x, \theta)$ without directly considering the underlying probabilities. This section serves as the foundation and motivation for our proposed method, which leverages uncertainty estimation to improve group robustness.
>
> Theorem 3.1 stands for the theoretical support of our approach. It demonstrates that estimating $\hat{p}(g | x, \theta)$ is feasible under the data generation process of $g$ and $(x, y)$ described in Eq. (3). While it is not explicitly referenced in Sections 4 and 5, Theorem 3.1 validates the core assumption underlying our method: that latent group probabilities can be inferred through uncertainty quantification. This theoretical result informs and justifies our design choices in the subsequent sections.
>
> Q2. *What are the evidence values for each class (the $e_k(x)$'s)?*
>
> **Response:**
>
> The evidence values $e_k(x)$ for each class $k$ are defined as the non-negative outputs of the neural network when processing the input $x$; specifically, the input is fed into the neural network $f_\theta(x)$, which produces outputs for each class, and these outputs are transformed into evidence by applying an activation function like ReLU or Softplus to ensure they are greater than or equal to zero—formally, $e_k(x) = \max(0, z_k(x))$ with ReLU or $e_k(x) = \ln(1 + e^{z_k(x)})$ with Softplus, where $z_k(x)$ are the pre-activation outputs (logits) of the network; thus, $e_k(x) \geq 0$ for all $k$, and these evidence values represent the support the model assigns to each class based on the input $x$.
>
> Q3. *What is $u(x_i)$ that appears on l.333?*
>
> **Response:** The term $u\left(x_i\right)$, as used on line 333 , represents the estimated posterior group probability for sample $x_i$. Formally, $u\left(x_i\right)$ is derived as:
>
> $$
> u\left(x_i\right)=\frac{K}{S\left(x_i\right)},
> $$
>
> where $K$ is the number of classes, and $S\left(x_i\right)=\sum_k \alpha_k\left(x_i\right)$ is the sum of the Dirichlet parameters. Intuitively, $u\left(x_i\right)$ quantifies the model's epistemic uncertainty for $x_i$, with higher values indicating greater uncertainty. This scalar is used during retraining to weight the loss contribution of the sample, prioritizing those with higher uncertainty for debiasing.
>
> Q4. *Does the proposed method require knowledge of the set of possible groups? (eg for waterbirds, does the method need to know that the set of possible groups are land bird-land bg, land bird water bg, water bird, land bg, and water bird-water bg)?*
>
> **Response:** Our method does not require prior knowledge of the specific group combinations (e.g., the exact group structure in Waterbirds). Instead, it operates on the assumption that the latent group structure can be inferred through uncertainty quantification. Notably, when the ERM model training converges, minority groups tend to exhibit higher uncertainty due to the model’s reliance on spurious correlations or insufficient representation of these groups in the training data. This property allows our approach to identify and address minority groups effectively and scalable to different datasets, even when the degrees of group imbalance are different.

---

### Official Review · Reviewer_PaJo · 2024-11-05

**Soundness:** 3
**Presentation:** 1
**Contribution:** 2
**Rating:** 5
**Confidence:** 2

**Summary:**

This paper presents a unifying framework through which they view group robustness algorithms that operate without group information. The framework, based on Empirical Bayes, first infers group information p(g|x,y) and then optimizes a conditional label distribution. The paper demonstrates that the method falling out of this interpretation performs well on a set of standard group robustness benchmarks.

**Strengths:**

The paper studies an important problem and uses a well-motivated statistical framework. On the practical side, the results are undoubtedly better than the baselines they consider, and the method seems very promising for improving group robustness.

**Weaknesses:**

I had trouble following the paper after Section 4. In particular, I did not understand the connection between the Empirical Bayes framework and "evidential deep learning," which the paper does not do enough to explain. I suggest that the authors dedicate more time to explaining the connection between the two concepts and illustrating how their perspective makes such an approach the obvious one.

I was also somewhat underwhelmed by the theoretical result, which seems to be a restatement of Tweedie's formula with the variables replaced using the context of the paper. The authors can improve this section by (a) highlighting what is new/different between their result and Tweedie's formula, and more importantly (b) experimentally verifying to what extent the assumptions of their theorem hold in practice.

Finally, the paper consistently claims to be more robust to hyperparameters than other methods, but I did not see a study of hyperparameter robustness in the main text or in the appendix. The introduced method does seem to require hyperparameters (e.g., the evidence regularization strength) that seem like they would be important, so I am not sure the authors do enough to back up the claim that their method is more robust to hyperparameters than baselines.

**Questions:**

None beyond the weaknesses above.

---

> ### Author Response · Authors · 2024-11-28
> **Responses to Reviewer PaJo**
>
> Thank you for your thoughtful feedback and valuable questions. We are delighted that you found our work to be well-motivated and promising for improving group robustness. Below, we address your questions in detail:
>
> W1. *Connection between the Empirical Bayes framework and evidential deep learning:*
>
> **Response:**  We sincerely appreciate your feedback and acknowledge the need to clarify the connection between the Empirical Bayes framework and evidential deep learning. The Empirical Bayes framework provides a theoretical foundation for estimating latent group probabilities $\hat{p}(g | x, \theta)$ using observed data and unifies previous methods, even though they were proposed from different perspectives. Evidential deep learning complements this by offering a practical mechanism to quantify uncertainty, which serves as an empirical prior in the uncertainty estimation process. The connection between the Empirical Bayes framework and evidential deep learning is that the Empirical Bayes framework motivated the use of evidential deep learning as a proxy to estimate $\hat{p}(g | x, \theta)$ in a scalable and data-driven manner.
>
> W2. *Regarding the theoretical results: (a) highlighting what is new/different between their result and Tweedie's formula, and (b) experimentally verifying to what extent the assumptions of their theorem hold in practice.*
>
> **Response:** While our result builds on Tweedie’s formula, the new contribution lies in adapting it to the specific problem of group robustness under the Empirical Bayes framework. In particular: Our adaptation focuses on estimating latent group probabilities $\hat{p}(g | x, \theta)$ in the context of biased data, which introduces unique challenges compared to the general setting of Tweedie’s formula. The theorem explicitly incorporates the relationship between latent group variables and data distributions through the lens of uncertainty quantification. This framing is critical for justifying the motivation of our uncertainty-based method and directly supports the design of the proposed method. While we have not explicitly tested the assumptions of the theorem, the effectiveness of our proposed algorithm provides empirical support. The improved performance across diverse datasets suggests that the inferred group probabilities, guided by uncertainty quantification, align with the theoretical formulation.
>
> W3. *Claim of hyperparameter robustness.*
>
> **Response:**
>
> We appreciate the reviewer’s observation and would like to clarify our claim regarding hyperparameter robustness. Unlike previous methods, which rely on tuning additional task-specific hyperparameters to achieve optimal performance, our approach does not introduce any new hyperparameters that require manual tuning beyond traditional ones such as learning rate or number of epochs. For example, Just Train Twice (JTT) requires specifying the upweighting parameter and early stopping epochs, while Learning from Failure (LfF) depends on a hyperparameter $q$ to control the degree of amplification. These task-specific hyperparameters often need to be manually selected for different datasets, adding complexity and reducing generalizability. In contrast, our method relies only on the evidence regularization strength $\lambda$ during training, which is pre-defined to anneal smoothly from 0 to 1 based on the ratio of the current epoch number to the annealing step. As shown in Tables 7 and 8, our approach does not require additional hyperparameters beyond the necessary ones (e.g., learning rate, number of epochs), making it inherently more robust and simpler to apply across diverse datasets.

---

### Author Response · Authors · 2024-12-02

Dear reviewers,

As the discussion period draws to a close, we wanted to kindly check if there are any remaining points you would like us to address. We have responded to all concerns raised and conducted additional experiments to strengthen our manuscript further. If no further comments are provided on our rebuttal, we will assume that all points of criticism have been adequately addressed.

We greatly appreciate your valuable feedback and are happy to provide any additional clarifications if needed.

Authors of Submission7986

---

### Meta-Review · Area_Chair_w8TN · 2024-12-16

**Metareview:**

The submission worked on an important problem (group DRO), but unfortunately the proposed method has several major technical issues. More critically, clarity is another major issue that made our reviewers difficult to appreciate its novelty and significance. Therefore, the overall quality is not good enough to make it an ICLR paper.

**Additional Comments On Reviewer Discussion:**

The rebuttal didn't address the concerns from the reviewers.

---

### Decision · Program_Chairs · 2025-01-22

Reject